# Effects of Linear Versus Changes of Direction Repeated Sprints on Intermittent High Intensity Running Performance in High-level Junior Football Players over an Entire Season: A Randomized Trial

**DOI:** 10.3390/sports7080189

**Published:** 2019-08-06

**Authors:** Edvard H Sagelv, Ivar Selnæs, Sigurd Pedersen, Svein Arne Pettersen, Morten B Randers, Boye Welde

**Affiliations:** 1School of Sport Sciences, Faculty of Health Sciences, UiT the Arctic University of Norway, 9037 Tromsø, Norway; 2Faculty of Education and Arts, Nord University, 7600 Levanger, Norway; 3Department of Sport Sciences and Clinical Biomechanics, Faculty of Health Sciences, University of Southern Denmark, 5230 Odense, Denmark

**Keywords:** soccer, repeated sprint, high intensity running

## Abstract

Background: Changes of direction (COD) repeated sprints (RSs) might have greater relevance to football than linear RSs. We aimed to compare the effects of linear and COD RSs on intermittent high intensity running (HIR) over an entire season. Methods: In total, 19 high-level male football players (16–19 years) randomly performed linear RSs or COD RSs twice a week during their competitive season over 22 weeks. Yo-Yo intermittent recovery test level 2 (Yo-Yo IR2), and 10- and 20-m sprint was assessed pre-, mid- (11 weeks), and post-intervention (22 weeks). Maximal oxygen uptake (VO_2max_) was assessed pre- and post-intervention. Results: There was no interaction effect (time x group) in Yo-Yo IR2 (*p* = 0.36, _p_η^2^ = 0.06) or sprint tests (10 m: *p* = 0.55, _p_η^2^ = 0.04, 20 m: *p* = 0.28 _p_η^2^ = 0.08), and no change differences between groups. There was a main effect of time for Yo-Yo IR2 (*p* = 0.002, _p_η^2^ = 0.31) but not in sprints or VO_2max_. Conclusion: Linear and COD RS exercise twice a week over 22 weeks equally improves intermittent HIR performance but does not improve sprint time or aerobic power in high-level junior football players. However, due to our two-armed intervention, we cannot exclude possible effects from other exercise components in the players’ exercise program.

## 1. Introduction

There are multiple physiological demands to elite football, which includes both aerobic and anaerobic energy contributions [1,2,3]. Among these, high aerobic power is crucial [3,4]. However, a threshold of ≥60 mL·kg^−1^·min^−1^ in maximal oxygen uptake (VO_2max_) seems to be sufficient to meet the aerobic demands in elite male football [2,3,5,6,7,8]. As a result, anaerobic exercise, such as high intensity running (HIR) and sprint ability, have gained more attention from coaches and researchers [1,9,10], and may be more applicable in differentiating elite- and sub-elite levels in football than aerobic power [6,9,10,11,12]. Although sprints compose <10% of the covered distance during a football match [13], they are often associated with decisive parts of the match [10] and goal scoring [14]. The sprints in football usually lasts 2 to 4 s [15,16], where 97% and 75% of the sprints are shorter than 30 and 10 m, respectively [17]. Moreover, HIR and sprint distance in elite football has increased considerably over the past decades [18,19]. Thus, in addition to fast sprints, the ability to execute repeated bouts of HIR and sprints is now considered one of the decisive factors for elite male football performance [1,2,11,20,21].

Although HIR and sprints are supramaximal and consequently result in high anaerobic energy contributions [2,3,6,22,23], the repetitive nature in football, with insufficient recovery duration between runs for full resynthesizing of phosphocreatine, leads to large aerobic energy contributions [24,25], which may lead to a lower running intensity [24]. 

Several exercise modes can be applied to improve intermittent HIR performance, including resistance exercise [26,27], power exercise [27], high intensity aerobic interval exercise [28,29], and small-sided games [28,30,31,32]. However, to improve intermittent HIR in football players, the most frequently applied exercise is linear repeated sprint (RS) exercises [24,26,29,33,34,35,36,37,38,39,40,41], which may be considered to primarily improve RSs ability. However, RSs also show additional improvements in intermittent HIR [28]. Furthermore, time motion analysis has revealed over 700 changes of direction (COD) at high velocities during a football match [42]. As exercise modalities specific to the sport may be even more relevant for sport performance [29,43,44], RSs including COD may be more applicable than linear RSs for improving intermittent HIR in football [9,45,46,47,48,49,50,51]. However, the evidence base for the effect of COD RS exercise on intermittent HIR performance is sparse. While it seems that COD RS exercise is effective for improving intermittent HIR performance in elite football players [51], inconsistent findings in youth sub-elite football players make the effect in football players at this level unclear [29,47,50,52].

Moreover, as football involves both technical, tactical, and physical demands, incorporating all relevant exercise aspects in players’ exercise schedules is challenging [1,44]. Thus, implementing RSs into the everyday exercise schedule may be an effective strategy to tax both the anaerobic and aerobic energy system [1,24,25,44,53]. Previous studies have reported positive effects of implementing RS exercises both outside [26,40,47,54,55] and inside [38,44,51,56] the competitive season. However, the duration of these studies are short (≤12 weeks) [41]. As intermittent HIR performance is shown to vary over an entire season [57], the applicability of short-term studies assessing RS exercise for implementation in the long-term basis over an entire football season is limited. Thus, the aim of this study was to compare the effect of linear and COD RS exercise on intermittent HIR performance in highly trained junior football players over an entire season. 

## 2. Materials and Methods

### 2.1. Design and Participants

In this randomized, two-parallel group trial, 19 high-level male junior football players aged 16 to 19 years from the same football team volunteered to participate. The team played at the highest junior level (Under 19) in Trøndelag football region. In addition to their team´s usual exercise sessions and matches, the players were randomly assigned to carry out RS exercise twice a week as either linear or COD RS, for 22 weeks during their competitive season. In the team, there were four full backs, four central backs, six central midfielders, four wide midfielders, and two strikers. In order to assure equal distribution of positions in both intervention groups, the randomization was stratified by playing position using Research Randomizer [58]. The goalkeepers were excluded from the study but were invited to perform one of the interventions of their own choice. The players’ age and anthropometric characteristics at baseline are shown in Table 1, for the linear RS and COD RS group, respectively. In the three pre-season months prior to the study, the players exercised on average 13 h per week, of which 8 h were football practice, 3 h aerobic exercise, and 2 h resistance exercise. All players were informed about the purpose of the study, and the possibility to withdraw from the study without providing any reason, verbally and in writing before providing written informed consent. For those who were under 18 years, both their parents/legal guardians and the player provided oral and written informed consent. This study was carried out with ethical standards for sports and exercise science in accordance with the Deceleration of Helsinki, and the Norwegian Social Sciences Data Services approved the study in addition to the storage of personal data (Approval reference number: 38437). Further approval from a regional ethics committee was not required for this study as per applicable institutional and national guidelines and regulations for sport and exercise science. 

### 2.2. Test Concepts and Instruments

Prior to the intervention trials, the players underwent a Yo-Yo intermittent recovery level 2 test (Yo-Yo IR2) [20] for determining intermittent HIR performance, 10 and 20 m sprint for evaluation of acceleration ability, and a treadmill test to exhaustion in the laboratory for the evaluation of VO_2max_. For logistical reasons, the players first performed the VO_2max_-test. Thereafter, in the following week, the players first performed the 10 and 20 m sprint tests on one day, and the Yo-Yo IR2 the day after. The players performed the post-tests after 22 weeks in the same order as the pre-tests. Additionally, the Yo-Yo IR2 and the 10 and 20 m sprint tests were also evaluated as mid-tests following 11 weeks in the intervention. The day prior to all tests, the players underwent a low intensity football practice and were instructed to avoid any strenuous exercise on their own. On all the testing days, the players were instructed to report to the field and laboratory tests well hydrated and properly fueled with nutrients, as they would do prior to a football match.

### 2.3. The Yo-Yo Intermittent Recovery Test Level 2

The Yo-Yo IR2 test was carried out in a gym on a parquet floor and with a stable temperature of 20 °C. The Yo-Yo IR tests can be considered similar to the more commonly known 20 m shuttle run by Léger et al. [59], however, it defers with a 10 s active recovery period between each 2 × 20 m run where the players walk for 5 m, turn 180°, and walk back to the starting line. The players run back and forth between the start and finish line at a progressively increasing speed controlled by audio bleeps from an audio file connected to speakers. There are two Yo-Yo tests, level 1 and 2, where level 1 starts at a lower speed and usually lasts longer than level 2. Level 2 taxes the anaerobic system to a larger extent then level 1 and was created to assess intermittent HIR in high-level athletes, whereas level 1 can also assess intermittent HIR in athletes of lower fitness levels [20]. The players performed the Yo-Yo IR2 in this present study. Prior to the test, the players jogged at a self-selected low intensity speed for 15 min, followed by four 40 m progressive sprints until 90% of their subjectively determined maximum speed. Thereafter, level 14 in the Yo-Yo IR level 1 was run for 4 min. Prior to performing the test, the players consumed 300 mL of water. In the Yo-Yo IR2 test, when the players failed twice to reach the finishing line before the bleep, the covered distance was recorded as the test result.

### 2.4. 10 and 20 m Sprint Test

The 10 and 20 m sprint tests were conducted inside, in the same gym as the Yo-Yo IR2 test. Following 15 min of running at a self-paced low intensity speed with four progressive sprints, the players performed a specialized warm-up, including knee raises, heel kicks, running sideways, backwards, and forward, and a final sprint burst. Following the consumption of water (300 mL), the players first performed the 10 m sprint test. The players started with one foot in front of the other, behind the starting line. Photocells (TC-Timer, Bower Timing System, Draper, UT, USA) were placed 30 cm from the starting line, 30 cm over the floor on poles, and 10 m further away (the finish line), 100 cm over the floor. The players started on their own initiative by breaking the laser beam, which measures the time to reaching the finish line by breaking the laser beam at the finish line. The players were instructed to perform the tests with maximal effort, but no verbal encouragement was given during the tests. The players were given three attempts each with a 1 min recovery period between each sprint, where the best result was recorded as their 10 m sprint time. Thereafter, the finish line was moved 10 m further away and the players performed three attempts of the 20 m sprint under similar conditions. The 1 min recovery time was considered sufficient as it was >20 times the sprint time in both tests. The players were instructed to use the same pair of shoes in the pre-, mid-, and post-tests to minimize differences in friction between the floor and their shoes.

### 2.5. Maximal Oxygen Uptake

Prior to the VO_2max_ test, the players’ weight and height were measured with a portable scale (Seca 876, Seca GmbH & Co. KG, Hamburg, Germany) and a stable stadiometer (Seca 217, Seca GmbH & Co. KG, Hamburg, Germany), respectively. The players performed the same warm up as prior to the Yo-Yo IR2 test (15 min self-paced low intensity run), followed by four 40 m progressive sprints until 80%, and four 40 m progressive sprints until 90%, of maximum speed. Thereafter, the players entered the treadmill (h/p cosmos quasar, h/p cosmos sports & medical gmbh, Nussdorf-Trainstein, Germany) and were connected to an ergospirometry mixing chamber system (Oxycon-Pro, Jaeger Instr., Hoechberg, Germany) with a two-way mouthpiece (Hans Rudolph 2700 Instr., Shawnee, KS, USA) for oxygen uptake (VO_2_) recordings, and were also equipped with a nose-clip and a heart rate (HR) monitor (Polar, RS800 HR monitor, Polar Electro Oy, Kempele, Finland). The Oxycon-Pro is shown to provide valid recordings over shorter and longer test periods in high-level athletes when compared against the Douglas bag method [60]. Prior to each test, the sensors were calibrated for oxygen (O_2_) and carbon dioxide (CO_2_) using known gas concentrations of 16.00% and 4.90%, respectively, as well as ambient air. The inspiratory flow volume was manually calibrated using a 3 L volume syringe (Calibration Syringe, series 5530, Hans Rudolph Instr.; Shawnee, KS, USA). Respiratory variables were recorded every 10 s and HR every 5 s. The treadmill inclination was set to 5% and the speed to 8 km·h^−1^ when starting the ramp protocol, with a 1 km·h^−1^ increase every 45 s until exhaustion. Furthermore, 15 s before each speed increase, the players were asked if they could sustain a 1 km·h^−1^ increase, where they communicated with a thumb up and down for yes and no, respectively. If they answered no, the players were encouraged to continue running on the treadmill until exhaustion at the actual speed. VO_2max_ was determined as reaching a respiratory exchange ratio (RER) ≥1.05 [61], and defined as the median of the three consecutive highest 10 s recordings. Maximum heart rate (HR_max_) was defined as the highest stable HR during the last minute of the test. 

### 2.6. Repeated Sprint Exercise Interventions

All players were familiar with linear RSs as part of their team’s conditioning program for the preseason preparations. Figure 1 illustrates the build-up of the two RS exercises. The linear RSs group performed 40 m linear sprints (A), whereas the COD RSs group sprinted a self-created COD RSs exercise of 30 m (B), determined by a pilot experiment of 11 players, where approximately the same time was used to sprint 30 m with COD as sprinting linearly for 40 m. The directional sprint court (B) started with; (1) a 5 m sprint forward before turning 180°, (2) running 5 m back to the starting line, and (3) thereafter turning another 180° followed by a 20 m linear sprint. Each RSs exercise session lasted 8 min in total and consisted of three sets of four RSs with 30 s recovery between each sprint, and 55 s recovery between each set. Each sprint lasted 5 to 6 s, and the players were instructed to sprint with maximal effort in each sprint. Except for the different RSs exercises, all the players carried out similar football, aerobic, and resistance exercise sessions during the season.

Table 2 shows the timeline of the study. During the intervention over the 22 weeks, a typical exercise week had in total 45, 150, and 300 min of low, moderate, and high intensity football exercise, respectively. The RS exercises were performed twice a week at the extension of two football sessions, on Friday and Sunday. During the season, if the team had two matches in one week, the RS exercises were performed only once in those weeks in order to avoid any perception of fatigue approaching the match days in the weeks with two matches. The team played 15 league matches, of which 2 weeks included two matches per week, and 11 weeks included one match per week. The team played no training matches in this period. 

The players had a summer break for three weeks in July, where they performed a pre-specified exercise program developed by the coach and the researchers. The program consisted of four sessions per week; all four sessions started with a 20 min general warm-up of low intensity self-paced jogging. Following the warm up, two sessions consisted of resistance exercises with their own body weight (push-ups, burpee jumps, abdominal crunches etc.) for 15 min and two sessions consisted of the same RS exercise as they were assigned to upon randomization. For the RS exercises in the summer break, the players used a stop watch (heart rate monitor with time monitoring, classical stop watch, stop watch in their smart phones etc.) to ensure similar recovery times between sprints as in the sessions led by the coach during the team’s sessions. Prior to their summer break, the players also performed the 10 and 20 m sprint test and the Yo-Yo IR2 test as mid-tests in week 11. Following the summer break, the second part of the in-season in the fall consisted of 8 weeks of the same football practices and RS exercises as prior to the summer break, before performing post-tests, resulting in a 22-week long intervention period. 

### 2.7. Statistical Analysis

Statistical Package for Social Sciences (SPSS, version 25, International Business Machine Cooperation, Armonk, NY, USA) was used to perform all statistical analyses (Appendix A). Originally, 20 players were invited to this study, however, one player could not complete the study due to a long-term injury not related to this study, and he was excluded from the final analyses. The Shapiro Wilk test confirmed all data did not deviate from a normal distribution. We performed a 2 (RS exercise condition) by 3 (pre-, mid-, and post-tests) repeated measure analysis of variance (ANOVA) to assess the change from pre-, mid-, and post-testing in the Yo-Yo IR2 test, and the 10 and 20 m sprint tests. Bonferroni-corrected follow-up tests were applied to detect differences from pre- to mid-, pre- to post-, and mid- to post-tests if we observed a main effect of time. Assumptions of sphericity and equality of error variance was confirmed for all tests (all *p* > 0.10). A univariate ANOVA on gain-scores (post-test–pre-test), with exercise intervention as an independent variable, was performed to evaluate pre- to post-test changes in VO_2max_. Here, the intercept was interpreted as the main effect of time. Estimates of the effect size were calculated as partial eta squared (_p_η^2^), where 0.01–0.05, 0.06–0.13, and ≥0.14 _p_η^2^ was considered as small, medium, and large effect sizes, respectively [62]. A 9% increase in the Yo-Yo IR2 test following six weeks of COD RS exercise during the competitive season in male professional football players has previously been reported [51]. With a 95% power and an alpha level of 0.05, the predetermined statistical power to observe a similar difference in the Yo-Yo IR2 needed five players in each group. The data are presented as mean ± standard deviation (SD) unless otherwise is stated.

## 3. Results

The descriptive characteristics of the players are shown in Table 1; there were no differences in baseline assessments between the two groups (all *p* > 0.26).

Figure 2 illustrates the performance in the Yo-Yo IR2 for pre-, mid-, and post-tests. There was no interaction effect of time x group (*p* = 0.36, _p_η^2^ = 0.06) or any difference in the change between the two groups in the Yo-Yo IR2 test (increase linear RS group: 7.7 ± 13.6%, increase COD RS group: 10.6 ± 8.4%, between-subjects effect: *p* = 0.70, _p_η^2^ = 0.01). Overall, the players increased their Yo-Yo IR2 performance by 9.1 ± 11.2% from pre- to mid- to post-test in the Yo-Yo IR2 (main effect of time *p* = 0.002, _p_η^2^ = 0.31), where the significant increase was observed from pre- to post-test (*p* = 0.009) and no differences was observed from pre- to mid-test (*p* = 0.052) or mid- to post-test (*p* = 0.23). 

Table 3 shows the pre-, mid-, and post-test results for the 10 and 20 m sprint, and the pre- and post-test results for VO_2max_. There were no interaction effects of time x group in sprint times (10 m: *p* = 0.55, _p_η^2^ = 0.04, 20 m: *p* = 0.28, _p_η^2^ = 0.08) and no difference was observed for changes in sprint times between the two groups (10 m: *p* = 0.66, _p_η^2^ = 0.01, 20 m: *p =* 0.27, _p_η^2^ = 0.08). Overall, there were no main effects of time in sprint improvements from pre-, mid-, and post-test in neither the 10 m (*p* = 0.70, _p_η^2^ = 0.02) nor the 20 m test (*p* = 0.43, _p_η^2^ = 0.05).

Finally, for VO_2max_, we did not observe any change differences between the two groups (*p* = 0.80, _p_η^2^ = 0.04) and consequently, there was no main effect of time for the change from pre- to post-test (−0.4 ± 2.6 mL·kg^−1^·min^−1^, *p* = 0.56, _p_η^2^ = 0.02) (Table 3).

## 4. Discussion

The main findings of this randomized trial of high-level junior football players who included linear or COD RS exercise twice a week over an entire season of 22 weeks were: (1) Improved intermittent HIR performance; (2) no differences in the effect of linear or COD RSs on intermittent HIR performance; and (3) no changes in linear sprint times and aerobic power neither within nor between the two groups. 

The players in the present study improved their performance in the Yo-Yo IR2 test. Although not always consistent [63], previous studies in football players report considerable improvements in the Yo-Yo IR2 following RS exercise [33,36,37,39,40,43,51,64], which is supported by the latest meta-analysis, which evaluates the effect of RS exercise on intermittent HIR assessed with the Yo-Yo IR2 test [41]. This present study’s finding further highlights the applicability of RS exercise for improving intermittent HIR performance.

However, the aim of the present study was to elucidate whether COD RS or linear RS exercise would result in superior improvements in intermittent HIR in football players over the course of an entire season; the present study showed no differences in increased distance in the Yo-Yo IR2 test between the two groups. However, although not statistically different, the largest improvement in the COD RS group was observed from pre- to mid-test (~7%), while the linear RS group experienced their largest improvements from mid- to post-test (~6%). As the RS exercise carried out with COD was unfamiliar for the COD RS group, the initial pre- to mid-test improvement may be attributed to the exercise being specific to the Yo-Yo IR2 test, as observed previously [43,44]. Additionally, as the players were familiar with linear RS exercise from their preseason preparations, the new stimuli of COD may also be the reason for this initial increase in the COD RS group. When adding the post-test results, it appears that these differences disappeared when both groups had performed RS consistently over 22 weeks throughout the season. This observation may be considered the overall seasonal effect of RS exercise. Thus, in order to derive the maximal potential of RS exercise for improving intermittent HIR performance, a longer intervention period may be needed. Nevertheless, due to no statistical differences, this interpretation is speculative. Future research may elucidate on our assumption of a statistical type 2-error.

Nevertheless, when comparing our results with other studies that evaluated the effect of linear and COD RS exercise on the Yo-Yo IR2 test, previous studies show inconsistent results, with one study reporting superior improvements [51] and others reporting no additional effect of COD RS exercise compared with linear RS exercise [47,50]. There may be some explanations for the inconsistent findings; first, the COD RS exercise performed in the present study and previous studies were of different movement patterns and volume [47,50,51]. Second, the inconsistent results may also be attributed to the diversity in the duration of the intervention periods between previous studies (≤12 weeks) [29,47,50,51] and our study (22 weeks). 

The players in our study did not improve their sprint performance in the 10 and 20 m sprint, which is consistent with one previous study in youth players [65]. However, the majority of studies exploring the effect of RS exercise on 10 and 20 m sprints in football players report considerable positive effects on sprint times [26,50,55,66,67], which is supported by the latest meta-analysis on the topic [41]. There may be multiple explanations for the inconsistent results; the different RS exercise volume (90–800 m), time during the season (pre-, during-, post-season), baseline characteristics (adolescent, junior, or adult; sex; professional vs. amateur; initial fitness level), and finally a different intervention duration in the present study (22 weeks) compared with previous studies (4–12 weeks) [41]. As an example, the included players in the present study sprinted faster over the 10 m (~1.70 s) than previously reported in male football players (1.79–1.90 s) [11]. Thus, the initial high fitness level of the included players in this study can be considered high. However, as testing procedures in sprint tests are shown to largely influence the outcome [68], comparisons of baseline values across studies should be done with caution. 

Intermittent HIR and sprints are considered supramaximal with high anaerobic energy contributions [2,3,6,22,23]. However, due the repetitive nature of football and insufficient recovery duration between runs, large aerobic contributions are also present [24,25]. Thus, one could expect improvements in aerobic power following RS exercise. However, the players in this study experienced no improvements in aerobic power following 22 weeks of RS exercise. This is in contrast to one previous study in comparable junior players [29] but consistent with one study in professional football players [51]. The players in our study displayed a higher baseline VO_2max_ compared with the junior players in the study by Ferrari Bravo et al. [29], which may explain the inconsistent results. However, the two aforementioned studies reported similar VO_2max_-values for their included players [29,51]. Thus, the substantially different weekly volume of RS exercise (Ferrari Bravo et al. [29]: 2 × 720 m, our study: 2 × 360 m), which potentially resulted in different aerobic energy contributions, seems to be the most plausible explanation for the inconsistent results.

The present study examined the effect of linear RSs and COD RSs during the competitive season. Previous studies assessed the effect during the pre- [50] and the in-season [47,51]. On the one hand, as physical capacities should be the focus during pre-season, in-season exercise is usually used for tactical exercise together with the maintenance of physical capacities [1]. Thus, the competitive season may not be optimal timing for assessment of the effect of any additional physical exercise mode. On the other hand, in order to allow sub-elite junior players to reach the physical capacities of elite players, extra physical stimulus during the in-season may be one way to reach the elite level. In fact, the covered distance in the Yo-Yo IR2 test at the baseline by the included players in our study (~888 m) can already be considered as elite level [21], which further highlights the initial fitness levels of the included players in this study. Considering the large effect of time found for the Yo-Yo IR2 test in this present study, one may speculate whether further structured RS exercise would allow these players to reach the covered distance for top elite-level players (~1047 m) [21]. Moreover, as most players play an entire football match, substitutes and unused players may suffer little to no physical stimulus during a football match. Thus, including some extra RS exercise may be crucial for those with low physical stimulus during matches.

### 4.1. Strengths

Previous studies that evaluated intermittent HIR performance in football players were mainly of a short duration (<12 weeks) [41]. As intermittent HIR performance is shown to vary throughout a football season [57], studies of short duration may have limited applicability for implantation in long-term cycles, for example, over an entire season. Our study intervention lasted 22 weeks over an entire in-season. Although RS exercise is generally found to elicit initial improvements in intermittent HIR performance, the results from our study indicate that the expected initial improvement may peak, where further RS exercise over 12 weeks may not lead to further improvements. However, it is unknown whether increasing the RS exercise volume following 12 weeks would allow for larger improvements in intermittent HIR performance. 

Furthermore, the included players in this study displayed a high initial fitness level; the covered distance in the Yo-Yo IR2 test can already be considered elite [21], the sprint time was faster than previously reported sprint times for male football players [11], and finally, the aerobic power was above the suggested threshold of 60 mL·kg^−1^·min^−1^ [2,3,5,6,7,8]. As our results were inconsistent with some previous studies, the high fitness level of the players in our study may suggest that players with such high fitness levels do not respond similarly to players of lower fitness levels. Hence, this may suggest that players of high fitness should have higher volume or different exercise stimuli to further improve their physical capabilities.

### 4.2. Limitations

Although our predetermined statistical power allowed for randomization into more than two groups, we only randomized the players into two groups performing either RS exercise as linear or with COD without any third group acting as controls performing only the in-season exercise schedule of the team with no additional RS exercise. For example, small-sided games are also shown to improve performance in the Yo-Yo IR2 test [28]. Thus, due to no control group in our study, we cannot completely rule out whether the observed improvement in the Yo-Yo IR2 test was due to the RS exercise or to other components in the team’s exercise schedule. Consequently, this should be regarded as the main limitation of our study. As these players are pursuing a professional career, they agreed to participate on the condition that they would be included in one of the intervention groups. Additionally, since the players are pursuing a professional career and this study was carried out over an entire season, one may discuss whether excluding high-level players from participating in extra exercise is ethically appropriate. Nevertheless, as repeated sprint exercise is proven effective for improving HIR performance [41], the aim of this study was to assess whether linear or COD RSs would elicit superior effects on HIR performance under the assumption that both groups would improve their HIR performance to some extent. 

We created our own COD RS exercise in order to match the sprint times to the linear RS exercise. As we did not adopt a previously used COD RS exercise, this limits the comparability to other studies. Moreover, the COD RS exercise involved 180° turns, which may not be completely specific to football performance, as turns can also be less or more than 180°. For example, a previous study included 90° turns to allow more football-specific turns [51]. Nevertheless, COD RSs with 180° turns have been adopted previously to evaluate intermittent HIR [29], and is at least specific to the Yo-Yo IR2 test.

Finally, the Yo-Yo IR2 is a well-established intermittent HIR test [20,21,69], which is associated with HIR during a football match [20], and also physiological measures of aerobic and anaerobic metabolism [20,69]. However, the clinically relevant improvement needed in the Yo-Yo IR2 test to observe an improvement in a measure of football match performance is unknown. Thus, studies examining the clinical relevance of the Yo-Yo IR2 test to improve actual football performance measures are warranted. Moreover, to our knowledge, no study has examined the effect of RSs on actual football performance, and such studies may elucidate interesting and highly applicable aspects of intermittent HIR and RS exercise for football match performance.

## 5. Conclusions

Including linear or COD RS exercise twice a week in the in-season exercise schedule over an entire season of 22 weeks equally improves intermittent HIR performance. However, there were no translational effects for improvements in linear sprint time or aerobic power. Due to our two-armed intervention with no additional control group, we cannot conclusively rule out that the observed improvements in intermittent HIR performance were due to the RS exercises or whether other exercise components may have contributed to the results. 

## Figures and Tables

**Figure 1 sports-07-00189-f001:**
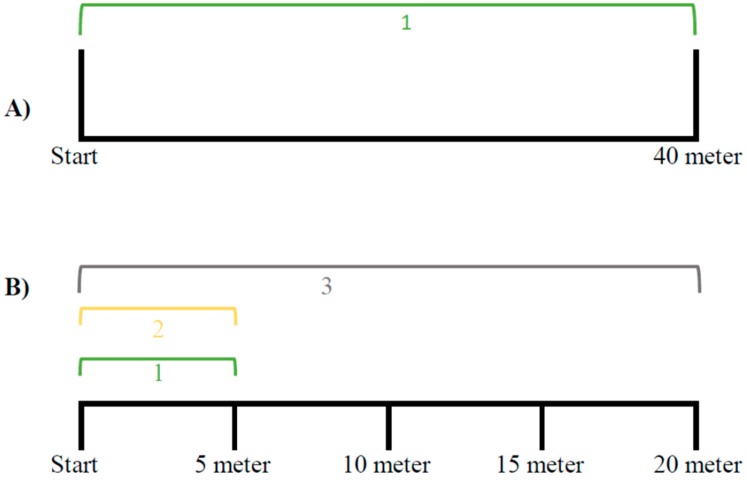
The two RSs exercises. (**A**) The linear RS group. (**B**) The COD RS group. RS = repeated sprint, COD = changes of direction.

**Figure 2 sports-07-00189-f002:**
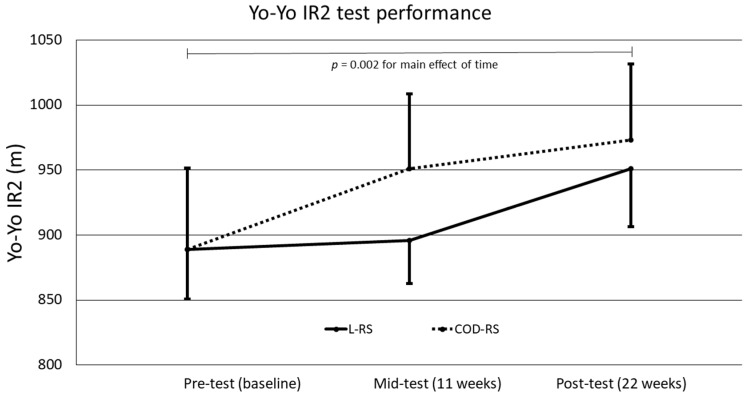
The Yo-Yo IR2 pre-, mid-, and post-test performance by linear RSs and the COD RSs group. Yo-Yo IR2 = Yo-Yo intermittent recovery test level 2, RS = repeated sprint, COD = changes of direction. Data are shown as mean and error bars are standard error of the mean.

**Table 1 sports-07-00189-t001:** Descriptive characteristics of the players when entering the study.

	Linear RS (n = 10)	COD RS (n = 9)
Age (year)	17.3 ± 0.5	17.4 ± 0.7
Weight (kg)	70.0 ± 7.6	71.3 ± 6.6
Height (m)	1.79 ± 0.05	1.80 ± 0.05
BMI (kg·m^−2^)	21.7 ± 2.4	21.9 ± 1.9

Data are shown as mean ± SD. COD = changes of direction, RS = repeated sprint, BMI = body mass index.

**Table 2 sports-07-00189-t002:** The timeline of the study.

	**Pre-Season**	**Pre-Tests**	**Spring Season**	
Duration	3 months	2 weeks	11 weeks	
Exercise content	13 h: 8 h football practice 3 h aerobic exercise 2 h resistance exercise	Week 1: VO_2max_ Week 2: 10 m 20 m Yo-Yo IR2	Typical week: Monday: Football practice, high intensity, low volume, 30 min Tuesday: Match Wednesday: Football practice, low intensity, 1.5 h Thursday: Football practice, high intensity, 1.5 h Friday: Large-sided simulated games, RS exercise, moderate intensity, 1.5 h Saturday: Aerobic (30 min running or bicycling) and resistance exercise (30 min body weight exercises), moderate intensity, 1 h Sunday: Small-sided games, high intensity, RS exercise, 1.5 h	
	**Mid-Tests**	**Summer Break**	**Fall Season**	**Post-Tests**
Duration	last week of spring season	3 weeks	8 weeks	2 weeks
Exercise content	10 m 20 m Yo-Yo IR2	4 sessions: 2 resistance exercise 2 RS exercise	Typical week: Monday: Football practice, high intensity, low volume, 30 min Tuesday: Match Wednesday: Football practice, low intensity, 1.5 h Thursday: Football practice, high intensity, 1.5 h Friday: Large-sided simulated games, RS exercise, moderate intensity, 1.5 h Saturday: Aerobic (30 min running or bicycling) and resistance exercise (30 min body weight exercises), moderate intensity, 1 h Sunday: Small-sided games, high intensity, RS exercise, 1.5 h	Week 1: VO_2max_ Week 2: 10 m 20 m Yo-Yo IR2

VO_2max_ = maximal oxygen uptake, Yo-Yo IR2: Yo-Yo intermittent recovery test level 2, RS = repeated sprints, 10 m = 10 m sprint, 20 m = 20 m sprint.

**Table 3 sports-07-00189-t003:** Pre-, mid- (11 weeks), and post-test (22 weeks) results of the 10 and 20 m sprint time, and pre- and post-test results for VO_2max_.

	Pre-Test (Baseline)		Mid-Test (11 Weeks)		Post-Test (22 Weeks)	
	L-RS (n = 10)	COD-RS (n = 9)	L-RS (n = 10)	COD-RS (n = 9)	L-RS (n = 10)	COD-RS (n = 9)
10 m (s)	1.68 ± 0.07	1.70 ± 0.07	1.68 ± 0.06	1.69 ± 0.07	1.69 ± 0.08	1.69 ± 0.06
20 m (s)	2.97 ± 0.10	3.05 ± 0.15	2.98 ± 0.10	3.05 ± 0.13	2.99 ± 0.11	3.03 ± 0.15
VO_2max_ (mL·kg^−1^·min^−1^)	62.6 ± 4.4	62.2 ± 6.1	N/A	N/A	62.4 ± 5.4	61.7 ± 5.1

L-RS = linear repeated sprint group, COD-RS = changes of direction repeated sprint group, Yo-Yo IR2 = Yo-Yo intermittent recovery test level 2, VO_2max_ = maximal oxygen uptake. Data are shown as mean ± SD.

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
