# Peer review of "Effects of Linear Versus Changes of Direction Repeated Sprints on Intermittent High Intensity Running Performance in High-level Junior Football Players over an Entire Season: A Randomized Trial"

_sports, 2019, doi:10.3390/sports7080189_

Round 1

Reviewer 1 Report

General comments:

The authors present a study about the effects of two methodologies of RS (lineal vs COD) in HIR in trained junior football players. The manuscript is well written and easily understood, but has serious methodological concerns:

§  The main problem is the lack of control of some important variables as the intensity football practice (the authors don’t provide any data when, I suppose, these data are the biggest training time percentage) and, aerobic and resistance training variables were self-administrated (the authors neither provide any data). For these reasons and taking into account that these variables are almost 95% of the total training time, I think that the results and conclusions should be regarded with caution.

Specific comments

The introduction provides sufficient background and includes relevant references

Materials and methods.

The authors describe the sample study as highly trained male junior football players, but they don´t provide data about their competition level. It’s strange that a football season for U19 players lasts 22 weeks (with a summer break of 3 weeks at 11 weeks to start the season). The players played the matches on Tuesday unless those were training matches (the authors don’t clarify this question). It’s still very strange that the study team only played 15 matches in a season of 22 weeks. The players had high VO2max values. I believe that is a mistake related to the protocol used.

Page 5, line 89…the authors really think that the study was a randomized controlled trial? Do the sample characteristics allow these experimental design?

Page 5, line 94…only 2 strikers in the team? The sample is unbalanced

Page 8, lines 171-174…Did the authors really calibrate the Oxycon-Pro prior to each test?

Page 9, lines 175-176…Can the authors reference this stress protocol?

Page 9, lines 180-181…Why the authors used this protocol to determine the VO2max?

Page 9, lines 187-189…Can the authors reference this COD RS protocol?. In my modest opinion, the training stimulus is inadequate and distant to the football reality.

Page 9, line 192…Why 55 seconds recovery between sets?

Page 10, line 203… aerobic and resistance exercise self-administrated. How and who controls these training?

Statistical analysis

Page 11, line 203…Why the authors use the partial eta square to estimate the effect size? This parameter is very detached from the sport sciences reality (read Hopkins 2009)

Discussion

In the discussion, all the arguments are focused on the differences among the duration of the previous studies and the training stimulus volume differences and, this is a very poor argument.

Conclusions

I don’t think that the conclusions are supported by the results.

In addition, the limitations are future research lines, not limitations

Reviewer 2 Report

I really enjoyed reading this and found it to be a thorough paper, answering many questions in the text that I may have asked.

Lines 58-61 I like the first paragraph, however, if this separate sentence could be included in this discussion in some way rather than separately it would further enhance this rationale.

Lines 207-208 – I don’t understand this. I get the purpose but cannot work out what this means.

Lines 234-235 – the table needs a number and title. Ok not sure how it appears here. In my copy the table is part way through the previous paragraph, perhaps just check this.

As this is a longitudinal study it may be best to insert a timeline showing the described sessions from the method. There is so much information to take in that even after re-reading several time it is unclear as to whether some of the training may have occurred in the off-season etc.  

Reviewer 3 Report

The aim of the reviewed study/article was to compare the effect of linear- and COD RS exercise on intermittent HIR performance in highly trained junior football players over an entire season.

The article has been divided into the chapters and subchapters correctly. The authors cite 69 references, 80% of which are articles published in the last 10 years. The research tools and statistical tests, despite a small research groups, have been selected properly.

The article takes up a very interesting and important issues. Despite the use of  tools and research methods used by the other Researchers, the Authors obtained interesting results important for their scientific discipline. The reviewed paper contains a several publishing errors described in the detailed notes.

Detailed comments:

 1. I suggest to improve the formatting of table and drawing descriptions:

Table 1 to which the authors refer on the page No. 5 in the chapter „Materials and Methods” – „Design and Participants” appears on the page No. 12 in the chapter "Statistical analysis". The description of the Table 1 is placed in the chapter „Results”; I suggest to place the Table 1 in the chapter „Materials and Methods” – „Design and Participants” in accordance with the magazine's guidelines regarding the formatting of tables and their descriptions

The authors refer to the Table 2 in the chapter "Results" (page No. 12), description of the Table 2 is on the page No. 13, but Table 2 does not appear at all; Please put the Table 2 in the "Results" chapter in accordance with the magazine's guidelines regarding the formatting of tables and their descriptions;

The Figure 1 and Figure 2 description need to be moved below the picture in accordance with the magazine's guidelines regarding the formatting of tables and their descriptions;

 Remove from the Figure 1 and Figure 2 description the sentence "Insert Figure ... about here:"

Remove from the Table 1 and Table 2 description the sentence "Insert Table ... about here:"

2. I suggest to present the VO2max values (for all tests) as a table or figure;

3. Limitations and strengths of your study need to be added into the „Discussion”     chapter.

Round 2

Reviewer 1 Report

I have read the comments of the authors, but I still think that the methodological concerns remain the same. Therefore I reaffirm my decision.

Author Response

Dear reviewer, 

As we cannot see your decision regarding endorsement of this manuscript, we can only assume, based on your comment, that you did not feel our answer and revisions as sufficient. We got some feedback from the editor based on your comments and the revised manscript. Without any further elucidation of our questions to you, we have now revised our manuscript according to the suggestion of the editor. Please see below the comments from the editor and our second revisions:

"1. Please verify in the paper that the conditions of a randomized controlled trial are met
2. Justify the use of COD RS protocol and let the reader know that this was self-created and perhaps list this limitation in the Discussion.
3. Please comment on how aerobic and resistance exercise was self-administrated in the paper (reader needs to know that)."

Our reply and revisions: 

1. After careful considerations, we agree that our study is not a controlled trial, as we did not include any third group that could act as controls. Accordingly, we have changed our title: it now says "randomized trial", as compared to previously where it said "randomized controlled trial". This is also changed in the beginning of our methods (Line 89. As this is more accurate, this is important to change. Thank you for highlighting the need to change this.

2. We have now stated in the methods that this was self created (Line 191), and this is justified by creating similar sprint times in the two RS protocols (Line 192-193). We have also included a paragraph on this under limitations, where we again highlight why we created an own COD RS protocol (Line 405-407).

3. Ok, this is now done (Line 213-221). The players were given a exercise program developed by the coach (resistance exercise sessions), and the self-administrated RS exercises was controlled by the players using a stop watch (either classical, through a heart rate watch with time monitoring, or their own smart phone). Indeed, this is also important and we apologize for not specifying this. 

4. Finally, we also included a final limitation to the COD RS protocol, included in the same paragraph were we mention that the COD RS protocol was self-created; this limitation addresses the issue raised by reviewer 1, where the COD RS only included 180 degrees turns, which may not be completely football specific. see line 407-411.

If you or the reviewers feel any more issues needs to be addressed, we are happy to further elucidate any concerns in our manuscript. The manuscript is now with tracked changes, and in the same file, a "clean version" without tracked changes can be found at the end.

Thank you and kind regards
Corresponding author Edvard H Sagelv

Reviewer 3 Report

The authors took into account the comments contained in the review and provided an comprehensive answer to the asked questions. In its current form the work is acceptable.

Author Response

Dear reviewer, 

Thank you again for reviewing our paper, and we are pleased to see that you are satisfied with our revisions and answers. 

As we cannot see your decision regarding endorsement of this manuscript, we can only assume, based on your comment, that you feel our answer and revisions as sufficient. However, we got some feedback from the editor based on the revised manuscript. If of interest, please see below the comments from the editor and our second revisions:

"1. Please verify in the paper that the conditions of a randomized controlled trial are met
2. Justify the use of COD RS protocol and let the reader know that this was self-created and perhaps list this limitation in the Discussion.
3. Please comment on how aerobic and resistance exercise was self-administrated in the paper (reader needs to know that)."

Our reply and revisions: 

1. After careful considerations, we agree that our study is not a controlled trial, as we did not include any third group that could act as controls. Accordingly, we have changed our title: it now says "randomized trial", as compared to previously where it said "randomized controlled trial". This is also changed in the beginning of our methods (Line 89. As this is more accurate, this is important to change. Thank you for highlighting the need to change this.

2. We have now stated in the methods that this was self created (Line 191), and this is justified by creating similar sprint times in the two RS protocols (Line 192-193). We have also included a paragraph on this under limitations, where we again highlight why we created an own COD RS protocol (Line 405-407).

3. Ok, this is now done (Line 213-221). The players were given a exercise program developed by the coach (resistance exercise sessions), and the self-administrated RS exercises was controlled by the players using a stop watch (either classical, through a heart rate watch with time monitoring, or their own smart phone). Indeed, this is also important and we apologize for not specifying this. 

4. Finally, we also included a final limitation to the COD RS protocol, included in the same paragraph were we mention that the COD RS protocol was self-created; this limitation addresses the issue raised by reviewer 1, where the COD RS only included 180 degrees turns, which may not be completely football specific. see line 407-411.

If you or the reviewers feel any more issues needs to be addressed, we are happy to further elucidate any concerns in our manuscript. The manuscript is now with tracked changes, and in the same file, a "clean version" without tracked changes can be found at the end.

Thank you and kind regards
Corresponding author Edvard H Sagelv